# Stochastic Structured Prediction under Bandit Feedback

**Artem Sokolov$^{\diamond,*}$, Julia Kreutzer$^*$, Christopher Lo$^{\dagger,*}$, Stefan Riezler$^{\ddagger,*}$**
$^*$Computational Linguistics & $^\ddagger$IWR, Heidelberg University, Germany
{sokolov,kreutzer,riezler}@cl.uni-heidelberg.de
$^\dagger$Department of Mathematics, Tufts University, Boston, MA, USA
chris.aa.lo@gmail.com
$^\diamond$Amazon Development Center, Berlin, Germany

## Abstract

Stochastic structured prediction under bandit feedback follows a learning protocol where on each of a sequence of iterations, the learner receives an input, predicts an output structure, and receives partial feedback in form of a task loss evaluation of the predicted structure. We present applications of this learning scenario to convex and non-convex objectives for structured prediction and analyze them as stochastic first-order methods. We present an experimental evaluation on problems of natural language processing over exponential output spaces, and compare convergence speed across different objectives under the practical criterion of optimal task performance on development data and the optimization-theoretic criterion of minimal squared gradient norm. Best results under both criteria are obtained for a non-convex objective for pairwise preference learning under bandit feedback.

## 1 Introduction

We present algorithms for stochastic structured prediction under bandit feedback that obey the following learning protocol: On each of a sequence of iterations, the learner receives an input, predicts an output structure, and receives partial feedback in form of a task loss evaluation of the predicted structure. In contrast to the full-information batch learning scenario, the gradient cannot be averaged over the complete input set. Furthermore, in contrast to standard stochastic learning, the correct output structure is not revealed to the learner. We present algorithms that use this feedback to "banditize" expected loss minimization approaches to structured prediction [18, 25]. The algorithms follow the structure of performing simultaneous exploration/exploitation by sampling output structures from a log-linear probability model, receiving feedback to the sampled structure, and conducting an update in the negative direction of an unbiased estimate of the gradient of the respective full information objective. The algorithms apply to situations where learning proceeds online on a sequence of inputs for which gold standard structures are not available, but feedback to predicted structures can be elicited from users. A practical example is interactive machine translation where instead of human generated reference translations only translation quality judgments on predicted translations are used for learning [20]. The example of machine translation showcases the complexity of the problem: For each input sentence, we receive feedback for only a single predicted translation out of a space that is exponential in sentence length, while the goal is to learn to predict the translation with the smallest loss under a complex evaluation metric.

[19] showed that partial feedback is indeed sufficient for optimization of feature-rich linear structured prediction over large output spaces in various natural language processing (NLP) tasks. Their experiments follow the standard online-to-batch conversion practice in NLP applications where the

---

$^*$ The work for this paper was done while the authors were at Heidelberg University.

model with optimal task performance on development data is selected for final evaluation on a test set. The contribution of our paper is to analyze these algorithms as stochastic first-order (SFO) methods in the framework of [7] and investigate the connection of optimization for task performance with optimization-theoretic concepts of convergence.

Our analysis starts with revisiting the approach to stochastic optimization of a non-convex expected loss criterion presented by [20]. The iteration complexity of stochastic optimization of a non-convex objective $J(w_t)$ can be analyzed in the framework of [7] as $\mathcal{O}(1/\epsilon^2)$ in terms of the number of iterations needed to reach an accuracy of $\epsilon$ for the criterion $\mathbb{E}[\|\nabla J(w_t)\|^2] \leq \epsilon$. [19] attempt to improve convergence speed by introducing a cross-entropy objective that can be seen as a (strong) convexification of the expected loss objective. The known best iteration complexity for strongly convex stochastic optimization is $\mathcal{O}(1/\epsilon)$ for the suboptimality criterion $\mathbb{E}[J(w_t)] - J(w^*) \leq \epsilon$. Lastly, we analyze the pairwise preference learning algorithm introduced by [19]. This algorithm can also be analyzed as an SFO method for non-convex optimization. To our knowledge, this is the first SFO approach to stochastic learning form pairwise comparison feedback, while related approaches fall into the area of gradient-free stochastic zeroth-order (SZO) approaches [24, 1, 7, 4]. Convergence rate for SZO methods depends on the dimensionality $d$ of the function to be evaluated, for example, the non-convex SZO algorithm of [7] has an iteration complexity of $\mathcal{O}(d/\epsilon^2)$. SFO algorithms do not depend on $d$ which is crucial if the dimensionality of the feature space is large as is common in structured prediction.

Furthermore, we present a comparison of empirical and theoretical convergence criteria for the NLP tasks of machine translation and noun-phrase chunking. We compare the empirical convergence criterion of optimal task performance on development data with the theoretically motivated criterion of minimal squared gradient norm. We find a correspondence of fastest convergence of pairwise preference learning on both tasks. Given the standard analysis of asymptotic complexity bounds, this result is surprising. An explanation can be given by measuring variance and Lipschitz constant of the stochastic gradient, which is smallest for pairwise preference learning and largest for cross-entropy minimization by several orders of magnitude. This offsets the possible gains in asymptotic convergence rates for strongly convex stochastic optimization, and makes pairwise preference learning an attractive method for fast optimization in practical interactive scenarios.

## 2  Related Work

The methods presented in this paper are related to various other machine learning problems where predictions over large output spaces have to be learned from partial information.

Reinforcement learning has the goal of maximizing the expected reward for choosing an action at a given state in a Markov Decision Process (MDP) model, where unknown rewards are received at each state, or once at the final state. The algorithms in this paper can be seen as one-state MDPs with context where choosing an action corresponds to predicting a structured output. Most closely related are recent applications of policy gradient methods to exponential output spaces in NLP problems [22, 3, 15]. Similar to our expected loss minimization approaches, these approaches are based on non-convex models, however, convergence rates are rarely a focus in the reinforcement learning literature. One focus of our paper is to present an analysis of asymptotic convergence and convergence rates of non-convex stochastic first-order methods.

Contextual one-state MDPs are also known as contextual bandits [11, 13] which operate in a scenario of maximizing the expected reward for selecting an arm of a multi-armed slot machine. Similar to our case, the feedback is partial, and the models consist of a single state. While bandit learning is mostly formalized as online regret minimization with respect to the best fixed arm in hindsight, we characterize our approach in an asymptotic convergence framework. Furthermore, our high-dimensional models predict structures over exponential output spaces. Since we aim to train these models in interaction with real users, we focus on the ease of elicitability of the feedback and on speed of convergence. In the spectrum of stochastic versus adversarial bandits, our approach is semi-adversarial in making stochastic assumptions on inputs, but not on rewards [12].

Pairwise preference learning has been studied in the full information supervised setting [8, 10, 6] where given preference pairs are assumed. Work on stochastic pairwise learning has been formalized as derivative-free stochastic zeroth-order optimization [24, 1, 7, 4]. To our knowledge, our approach

---

**Algorithm 1** Bandit Structured Prediction

1: Input: sequence of learning rates $\gamma_t$
2: Initialize $w_0$
3: **for** $t = 0, \ldots, T$ **do**
4:     Observe $x_t$
5:     Sample $\tilde{y}_t \sim p_{w_t}(y|x_t)$
6:     Obtain feedback $\Delta(\tilde{y}_t)$
7:     $w_{t+1} = w_t - \gamma_t \, s_t$
8: Choose a solution $\hat{w}$ from the list $\{w_0, \ldots, w_T\}$

---

to pairwise preference learning from partial feedback is the first SFO approach to learning from pairwise preferences in form of relative task loss evaluations.

## 3 Expected Loss Minimization for Structured Prediction

[18, 25] introduce the expected loss criterion for structured prediction as the minimization of the expectation of a given task loss function with respect to the conditional distribution over structured outputs. Let $\mathcal{X}$ be a structured input space, let $\mathcal{Y}(x)$ be the set of possible output structures for input $x$, and let $\Delta_y : \mathcal{Y} \to [0,1]$ quantify the loss $\Delta_y(y')$ suffered for predicting $y'$ instead of the gold standard structure $y$. In the full information setting, for a given (empirical) data distribution $p(x,y)$, the learning problem is defined as

$$\min_{w \in \mathbb{R}^d} \mathbb{E}_{p(x,y)p_w(y'|x)} \left[ \Delta_y(y') \right] = \min_{w \in \mathbb{R}^d} \sum_{x,y} p(x,y) \sum_{y' \in \mathcal{Y}(x)} \Delta_y(y') p_w(y'|x), \tag{1}$$

where

$$p_w(y|x) = \exp(w^\top \phi(x,y))/Z_w(x) \tag{2}$$

is a Gibbs distribution with joint feature representation $\phi : \mathcal{X} \times \mathcal{Y} \to \mathbb{R}^d$, weight vector $w \in \mathbb{R}^d$, and normalization constant $Z_w(x)$. Despite being a highly non-convex optimization problem, positive results have been obtained by gradient-based optimization with respect to

$$\nabla \mathbb{E}_{p(x,y)p_w(y'|x)} \left[ \Delta_y(y') \right] = \mathbb{E}_{p(x,y)p_w(y'|x)} \left[ \Delta_y(y') \left( \phi(x,y') - \mathbb{E}_{p_w(y'|x)}[\phi(x,y')] \right) \right]. \tag{3}$$

Unlike in the full information scenario, in structured learning under bandit feedback the gold standard output structure $y$ with respect to which the objective function is evaluated is not revealed to the learner. Thus we can neither evaluate the task loss $\Delta$ nor calculate the gradient (3) as in the full information case. A solution to this problem is to pass the evaluation of the loss function to the user, i.e, we access the loss directly through user feedback without assuming existence of a fixed reference $y$. In the following, we will drop the subscript referring to the gold standard structure in the definition of $\Delta$ to indicate that the feedback is in general independent of gold standard outputs. In particular, we allow $\Delta$ to be equal to 0 for several outputs.

## 4 Stochastic Structured Prediction under Partial Feedback

**Algorithm Structure.** Algorithm 1 shows the structure of the methods analyzed in this paper. It assumes a sequence of input structures $x_t, t = 0, \ldots, T$ that are generated by a fixed, unknown distribution $p(x)$ (line 4). For each randomly chosen input, an output $\tilde{y}_t$ is sampled from a Gibbs model to perform simultaneous exploitation (use the current best estimate) / exploration (get new information) on output structures (line 5). Then, feedback $\Delta(\tilde{y}_t)$ to the predicted structure is obtained (line 6). An update is performed by taking a step in the negative direction of the stochastic gradient $s_t$, at a rate $\gamma_t$ (line 7). As a post-optimization step, a solution $\hat{w}$ is chosen from the list of vectors $w_t \in \{w_0, \ldots, w_T\}$ (line 8).

Given Algorithm 1, we can formalize the notion of "banditization" of objective functions by presenting different instantiations of the vector $s_t$, and showing them to be unbiased estimates of the gradients of corresponding full information objectives.

**Expected Loss Minimization.**   [20] presented an algorithm that minimizes the following expected loss objective. It is non-convex for the specific instantiations in this paper:

$$\mathbb{E}_{p(x)p_w(y|x)}\left[\Delta(y)\right] = \sum_x p(x) \sum_{y \in \mathcal{Y}(x)} \Delta(y)p_w(y|x). \tag{4}$$

The vector $s_t$ used in their algorithm can be seen as a stochastic gradient of this objective, i.e., an evaluation of the full gradient at a randomly chosen input $x_t$ and output $\tilde{y}_t$:

$$s_t = \Delta(\tilde{y}_t)\left(\phi(x_t, \tilde{y}_t) - \mathbb{E}_{p_{w_t}(y|x_t)}[\phi(x_t, y)]\right). \tag{5}$$

Instantiating $s_t$ in Algorithm 1 to the stochastic gradient in equation (5) yields an update that compares the sampled feature vector to the average feature vector, and performs a step into the opposite direction of this difference, the more so the higher the loss of the sampled structure is. In the following, we refer to the algorithm for expected loss minimization defined by the update (5) as Algorithm *EL*.

**Pairwise Preference Learning.**   Decomposing complex problems into a series of pairwise comparisons has been shown to be advantageous for human decision making [23]. For the example of machine translation, this means that instead of requiring numerical assessments of translation quality from human users, only a relative preference judgement on a pair of translations needs to be elicited. This idea is formalized in [19] as an expected loss objective with respect to a conditional distribution of pairs of structured outputs. Let $\mathcal{P}(x) = \{\langle y_i, y_j \rangle | y_i, y_j \in \mathcal{Y}(x)\}$ denote the set of output pairs for an input $x$, and let $\Delta(\langle y_i, y_j \rangle) : \mathcal{P}(x) \to [0, 1]$ denote a task loss function that specifies a dispreference of $y_i$ compared to $y_j$. In the experiments reported in this paper, we simulate two types of pairwise feedback. Firstly, continuous pairwise feedback is computed as

$$\Delta(\langle y_i, y_j \rangle) = \begin{cases} \Delta(y_i) - \Delta(y_j) & \text{if } \Delta(y_i) > \Delta(y_j), \\ 0 & \text{otherwise.} \end{cases} \tag{6}$$

A binary feedback function is computed as

$$\Delta(\langle y_i, y_j \rangle) = \begin{cases} 1 & \text{if } \Delta(y_i) > \Delta(y_j), \\ 0 & \text{otherwise.} \end{cases} \tag{7}$$

Furthermore, we assume a feature representation $\phi(x, \langle y_i, y_j \rangle) = \phi(x, y_i) - \phi(x, y_j)$ and a Gibbs model on pairs of output structures

$$p_w(\langle y_i, y_j \rangle | x) = \frac{e^{w^\top(\phi(x,y_i) - \phi(x,y_j))}}{\sum\limits_{\langle y_i, y_j \rangle \in \mathcal{P}(x)} e^{w^\top(\phi(x,y_i) - \phi(x,y_j))}} = p_w(y_i|x)p_{-w}(y_j|x). \tag{8}$$

The factorization of this model into the product $p_w(y_i|x)p_{-w}(y_j|x)$ allows efficient sampling and calculation of expectations. Instantiating objective (4) to the case of pairs of output structures defines the following objective that is again non-convex in the use cases in this paper:

$$\mathbb{E}_{p(x)p_w(\langle y_i, y_j \rangle | x)}\left[\Delta(\langle y_i, y_j \rangle)\right] = \sum_x p(x) \sum_{\langle y_i, y_j \rangle \in \mathcal{P}(x)} \Delta(\langle y_i, y_j \rangle)\, p_w(\langle y_i, y_j \rangle | x). \tag{9}$$

Learning from partial feedback on pairwise preferences will ensure that the model finds a ranking function that assigns low probabilities to discordant pairs with respect the the observed preference pairs. Stronger assumptions on the learned ranking can be made if asymmetry and transitivity of the observed ordering of pairs is required.[2] An algorithm for pairwise preference learning can be defined by instantiating Algorithm 1 to sampling output pairs $\langle \tilde{y}_i, \tilde{y}_j \rangle_t$, receiving feedback $\Delta(\langle \tilde{y}_i, \tilde{y}_j \rangle_t)$, and performing a stochastic gradient update using

$$s_t = \Delta(\langle \tilde{y}_i, \tilde{y}_j \rangle_t)\left(\phi(x_t, \langle \tilde{y}_i, \tilde{y}_j \rangle_t) - \mathbb{E}_{p_{w_t}(\langle y_i, y_j \rangle | x_t)}[\phi(x_t, \langle y_i, y_j \rangle)]\right). \tag{10}$$

The algorithms for pairwise preference ranking defined by update (10) are referred to as Algorithms *PR(bin)* and *PR(cont)*, depending on the use of binary or continuous feedback.

**Cross-Entropy Minimization.** The standard theory of stochastic optimization predicts considerable improvements in convergence speed depending on the functional form of the objective. This motivated the formalization of a convex upper bounds on expected normalized loss in [19]. If a normalized gain function $\bar{g}(y) = \frac{g(y)}{Z_g(x)}$ is used where $Z_g(x) = \sum_{y \in \mathcal{Y}(x)} g(y)$, and $g = 1 - \Delta$, the objective can be seen as the cross-entropy of model $p_w(y|x)$ with respect to $\bar{g}(y)$:

$$\mathbb{E}_{p(x)\bar{g}(y)}\left[-\log p_w(y|x)\right] = -\sum_x p(x) \sum_{y \in \mathcal{Y}(x)} \bar{g}(y) \log p_w(y|x). \tag{11}$$

For a proper probability distribution $\bar{g}(y)$, an application of Jensen's inequality to the convex negative logarithm function shows that objective (11) is a convex upper bound on objective (4). However, normalizing the gain function is prohibitive in a partial feedback setting since it would require to elicit user feedback for each structure in the output space. [19] thus work with an unnormalized gain function $g(y)$ that preserves convexity. This can be seen by rewriting the objective as the sum of a linear and a convex function in $w$:

$$\mathbb{E}_{p(x)g(y)}\left[-\log p_w(y|x)\right] = -\sum_x p(x) \sum_{y \in \mathcal{Y}(x)} g(y) w^\top \phi(x, y) \tag{12}$$
$$+ \sum_x p(x)(\log \sum_{y \in \mathcal{Y}(x)} \exp(w^\top \phi(x, y)))\alpha(x),$$

where $\alpha(x) = \sum_{y \in \mathcal{Y}(x)} g(y)$ is a constant factor not depending on $w$. Instantiating Algorithm 1 to the following stochastic gradient $s_t$ of this objective yields an algorithm for cross-entropy minimization:

$$s_t = \frac{g(\tilde{y}_t)}{p_{w_t}(\tilde{y}_t|x_t)} \left(-\phi(x_t, \tilde{y}_t) + \mathbb{E}_{p_{w_t}}[\phi(x_t, y_t)]\right). \tag{13}$$

Note that the ability to sample structures from $p_{w_t}(\tilde{y}_t|x_t)$ comes at the price of having to normalize $s_t$ by $1/p_{w_t}(\tilde{y}_t|x_t)$. While minimization of this objective will assign high probabilities to structures with high gain, as desired, each update is affected by a probability that changes over time and is unreliable when training is started. This further increases the variance already present in stochastic optimization. We deal with this problem by clipping too small sampling probabilities to $\hat{p}_{w_t}(\tilde{y}_t|x_t) = \max\{p_{w_t}(\tilde{y}_t|x_t), k\}$ for a constant $k$ [9]. The algorithm for cross-entropy minimization based on the stochastic gradient (13) is referred to as Algorithm *CE* in the following.

## 5 Convergence Analysis

To analyze convergence, we describe Algorithms *EL*, *PR*, and *CE* as stochastic first-order (SFO) methods in the framework of [7]. We assume lower bounded, differentiable objective functions $J(w)$ with Lipschitz continuous gradient $\nabla J(w)$ satisfying

$$\|\nabla J(w + w') - \nabla J(w)\| \leq L\|w'\| \quad \forall w, w', \exists L \geq 0. \tag{14}$$

For an iterative process of the form $w_{t+1} = w_t - \gamma_t s_t$, the conditions to be met concern unbiasedness of the gradient estimate

$$\mathbb{E}[s_t] = \nabla J(w_t), \quad \forall t \geq 0, \tag{15}$$

and boundedness of the variance of the stochastic gradient

$$\mathbb{E}[\|s_t - \nabla J(w_t)\|^2] \leq \sigma^2, \quad \forall t \geq 0. \tag{16}$$

Condition (15) is met for all three Algorithms by taking expectations over all sources of randomness, i.e., over random inputs and output structures. Assuming $\|\phi(x, y)\| \leq R$, $\Delta(y) \in [0, 1]$ and $g(y) \in [0, 1]$ for all $x, y$, and since the ratio $\frac{g(\tilde{y}_t)}{\hat{p}_{w_t}(\tilde{y}_t|x_t)}$ is bounded, the variance in condition (16) is bounded. Note that the analysis of [7] justifies the use of constant learning rates $\gamma_t = \gamma, t = 0, \ldots, T$.

Convergence speed can be quantified in terms of the number of iterations needed to reach an accuracy of $\epsilon$ for a gradient-based criterion $\mathbb{E}[\|\nabla J(w_t)\|^2] \leq \epsilon$. For stochastic optimization of non-convex objectives, the iteration complexity with respect to this criterion is analyzed as $\mathcal{O}(1/\epsilon^2)$ in [7]. This complexity result applies to our Algorithms *EL* and *PR*.

The iteration complexity of stochastic optimization of (strongly) convex objectives has been analyzed as at best $\mathcal{O}(1/\epsilon)$ for the suboptimality criterion $\mathbb{E}[J(w_t)] - J(w^*) \leq \epsilon$ for decreasing learning rates [14].[3] Strong convexity of objective (12) can be achieved easily by adding an $\ell_2$ regularizer $\frac{\lambda}{2}\|w\|^2$ with constant $\lambda > 0$. Algorithm *CE* is then modified to use the following regularized update rule $w_{t+1} = w_t - \gamma_t \left(s_t + \frac{\lambda}{T} w_t\right)$.

This standard analysis shows two interesting points: First, Algorithms *EL* and *PR* can be analyzed as SFO methods where the latter only requires relative preference feedback for learning, while enjoying an iteration complexity that does not depend on the dimensionality of the function as in gradient-free stochastic zeroth-order (SZO) approaches. Second, the standard asymptotic complexity bound of $\mathcal{O}(1/\epsilon^2)$ for non-convex stochastic optimization hides the constants $L$ and $\sigma^2$ in which iteration complexity increases linearly. As we will show, these constants have a substantial influence, possibly offsetting the advantages in asymptotic convergence speed of Algorithm *CE*.

## 6  Experiments

**Measuring Numerical Convergence and Task Loss Performance.**   In the following, we will present an experimental evaluation for two complex structured prediction tasks from the area of NLP, namely statistical machine translation and noun phrase chunking. Both tasks involve dynamic programming over exponential output spaces, large sparse feature spaces, and non-linear non-decomposable task loss metrics. Training for both tasks was done by simulating bandit feedback by evaluating $\Delta$ against gold standard structures which are never revealed to the learner. We compare the empirical convergence criterion of optimal task performance on development data with numerical results on theoretically motivated convergence criteria.

For the purpose of measuring convergence with respect to optimal task performance, we report an evaluation of convergence speed on a fixed set of unseen data as performed in [19]. This instantiates the selection criterion in line (8) in Algorithm 1 to an evaluation of the respective task loss function $\Delta(\hat{y}_{w_t}(x))$ under MAP prediction $\hat{y}_w(x) = \arg\max_{y \in \mathcal{Y}(x)} p_w(y|x)$ on the development data. This corresponds to the standard practice of online-to-batch conversion where the model selected on the development data is used for final evaluation on a further unseen test set. For bandit structured prediction algorithms, final results are averaged over three runs with different random seeds.

For the purpose of obtaining numerical results on convergence speed, we compute estimates of the expected squared gradient norm $\mathbb{E}[\|\nabla J(w_t)\|^2]$, the Lipschitz constant $L$ and the variance $\sigma^2$ in which the asymptotic bounds on iteration complexity grow linearly.[4] We estimate the squared gradient norm by the squared norm of the stochastic gradient $\|s_T\|^2$ at a fixed time horizon $T$. The Lipschitz constant $L$ in equation (14) is estimated by $\max_{i,j} \frac{\|s_i - s_j\|}{\|w_i - w_j\|}$ for 500 pairs $w_i$ and $w_j$ randomly drawn from the weights produced during training. The variance $\sigma^2$ in equation (16) is computed as the empirical variance of the stochastic gradient, taken at regular intervals after each epoch of size $D$, yielding the estimate $\frac{1}{K}\sum_{k=1}^{K}\|s_{kD} - \frac{1}{K}\sum_{k=1}^{K} s_{kD}\|^2$ where $K = \lfloor \frac{T}{D} \rfloor$. All estimates include multiplication of the stochastic gradient with the learning rate. For comparability of results across different algorithms, we use the same $T$ and the same constant learning rates for all algorithms.[5]

**Statistical Machine Translation.**   In this experiment, an interactive machine translation scenario is simulated where a given machine translation system is adapted to user style and domain based on feedback to predicted translations. Domain adaptation from Europarl to NewsCommentary domains using the data provided at the WMT 2007 shared task is performed for French-to-English translation.[6]

The MT experiments are based on the synchronous context-free grammar decoder `cdec` [5]. The models use a standard set of dense and lexicalized sparse features, including an out-of and an in-

| Task | Algorithm | Iterations | Score | $\gamma$ | $\lambda$ | $k$ |
|---|---|---|---|---|---|---|
| SMT | **CE** | 281k | $0.271_{\pm 0.001}$ | 1e-6 | 1e-6 | 5e-3 |
| | **EL** | 370k | $0.267_{\pm 8e-6}$ | 1e-5 | | |
| | **PR**(bin) | 115k | $0.273_{\pm 0.0005}$ | 1e-4 | | |
| Chunking | **CE** | 5.9M | $0.891_{\pm 0.005}$ | 1e-6 | 1e-6 | 1e-2 |
| | **EL** | 7.5M | $0.923_{\pm 0.002}$ | 1e-4 | | |
| | **PR**(cont) | 4.7M | $0.914_{\pm 0.002}$ | 1e-4 | | |

Table 1: Test set evaluation for stochastic learning under bandit feedback from [19], for chunking under F1-score, and for machine translation under BLEU. Higher is better for both scores. Results for stochastic learners are averaged over three runs of each algorithm, with standard deviation shown in subscripts. The meta-parameter settings were determined on *dev* sets for constant learning rate $\gamma$, clipping constant $k$, $\ell_2$ regularization constant $\lambda$.

domain language model. The out-of-domain baseline model has around 200k active features. The pre-processing, data splits, feature sets and tuning strategies are described in detail in [19]. The difference in the task loss evaluation between out-of-domain (BLEU: 0.2651) and in-domain (BLEU: 0.2831) models gives the range of possible improvements (1.8 BLEU points) for bandit learning.

Learning under *bandit feedback* starts at the learned weights of the out-of-domain median models. It uses parallel in-domain data (`news-commentary`, 40,444 sentences) to simulate bandit feedback, by evaluating the sampled translation against the reference using as loss function $\Delta$ a smoothed per-sentence $1 - $ BLEU (zero $n$-gram counts being replaced with 0.01). After each update, the hypergraph is re-decoded and all hypotheses are re-ranked. Training is distributed across 38 shards using a multitask-based feature selection algorithm [17].

**Noun-phrase Chunking.** The experimental setting for chunking is the same as in [19]. Following [16], conditional random fields (CRF) are applied to the noun phrase chunking task on the CoNLL-2000 dataset[7]. The implemented set of feature templates is a simplified version of [16] and leads to around 2M active features. Training under full information with a log-likelihood objective yields 0.935 F1. In difference to machine translation, training with bandit feedback starts from $w_0 = \mathbf{0}$, not from a pre-trained model.

**Task Loss Evaluation.** Table 1 lists the results of the task loss evaluation for machine translation and chunking as performed in [19], together with the optimal meta-parameters and the number of iterations needed to find an optimal result on the development set. Note that the pairwise feedback type (*cont* or *bin*) is treated as a meta-parameter for Algorithm *PR* in our simulation experiment. We found that *bin* is preferable for machine translation and *cont* for chunking in order to obtain the highest task scores.

For machine translation, all bandit learning runs show significant improvements in BLEU score over the out-of-domain baseline. Early stopping by task performance on the development led to the selection of algorithm *PR(bin)* at a number of iterations that is by a factor of 2-4 smaller compared to Algorithms *EL* and *CE*.

For the chunking experiment, the F1-score results obtained for bandit learning are close to the full-information baseline. The number of iterations needed to find an optimal result on the development set is smallest for Algorithm *PR(cont)*, compared to Algorithms *EL* and *CE*. However, the best F1-score is obtained by Algorithm *EL*.

**Numerical Convergence Results.** Estimates of $\mathbb{E}[\|\nabla J(w_t)\|^2]$, $L$ and $\sigma^2$ for three runs of each algorithm and task with different random seeds are listed in Table 2.

For machine translation, at time horizon $T$, the estimated squared gradient norm for Algorithm *PR* is several orders of magnitude smaller than for Algorithms *EL* and *CE*. Furthermore, the estimated Lipschitz constant $L$ and the estimated variance $\sigma^2$ are smallest for Algorithm *PR*. Since the iteration complexity increases linearly with respect to these terms, smaller constants $L$ and $\sigma^2$ and a smaller

| Task | Algorithm | $T$ | $\|s_T\|^2$ | $L$ | $\sigma^2$ |
|------|-----------|-----|-------------|-----|------------|
| SMT | **CE** | 767,000 | $3.04_{\pm 0.02}$ | $0.54_{\pm 0.3}$ | $35_{\pm 6}$ |
| | **EL** | 767,000 | $0.02_{\pm 0.03}$ | $1.63_{\pm 0.67}$ | $3.13\text{e-}4_{\pm 3.60e-6}$ |
| | **PR**(bin) | 767,000 | $2.88\text{e-}4_{\pm 3.40e-6}$ | $0.08_{\pm 0.01}$ | $3.79\text{e-}5_{\pm 9.50e-8}$ |
| | **PR**(cont) | 767,000 | $1.03\text{e-}8_{\pm 2.91e-10}$ | $0.10_{\pm 5.70e-3}$ | $1.78\text{e-}7_{\pm 1.45e-10}$ |
| Chunking | **CE** | 3,174,400 | $4.20_{\pm 0.71}$ | $1.60_{\pm 0.11}$ | $4.88_{\pm 0.07}$ |
| | **EL** | 3,174,400 | $1.21\text{e-}3_{\pm 1.1e-4}$ | $1.16_{\pm 0.31}$ | $0.01_{\pm 9.51e-5}$ |
| | **PR**(bin) | 3,174,400 | $7.71\text{e-}4_{\pm 2.53e-4}$ | $1.33_{\pm 0.24}$ | $4.44\text{e-}3_{\pm 2.66e-5}$ |
| | **PR**(cont) | 3,174,400 | $5.99\text{e-}3_{\pm 7.24e-4}$ | $1.11_{\pm 0.30}$ | $0.03_{\pm 4.68e-4}$ |

Table 2: Estimates of squared gradient norm $\|s_T\|^2$, Lipschitz constant $L$ and variance $\sigma^2$ of stochastic gradient (including multiplication with learning rate) for fixed time horizon $T$ and constant learning rates $\gamma = 1e - 6$ for SMT and for chunking. The clipping and regularization parameters for *CE* are set as in Table 1 for machine translation, except for chunking *CE* $\lambda = 1e - 5$. Results are averaged over three runs of each algorithm, with standard deviation shown in subscripts.

value of the estimate $\mathbb{E}[\|\nabla J(w_t)\|^2]$ at the same number of iterations indicates fastest convergence for Algorithm *PR*. This theoretically motivated result is consistent with the practical convergence criterion of early stopping on development data: Algorithm *PR* which yields the smallest squared gradient norm at time horizon $T$ also needs the smallest number of iterations to achieve optimal performance on the development set. In the case of machine translation, Algorithm *PR* even achieves the nominally best BLEU score on test data.

For the chunking experiment, after $T$ iterations, the estimated squared gradient norm and either of the constants $L$ and $\sigma^2$ for Algorithm *PR* are several orders of magnitude smaller than for Algorithm *CE*, but similar to the results for Algorithm *EL*. The corresponding iteration counts determined by early stopping on development data show an improvement of Algorithm *PR* over Algorithms *CE* and *EL*, however, by a smaller factor than in the machine translation experiment.

Note that for comparability across algorithms, the same constant learning rates were used in all runs. However, we obtained similar relations between algorithms by using the meta-parameter settings chosen on development data as shown in Table 1. Furthermore, the above tendencies hold for both settings of the meta-parameter *bin* or *cont* of Algorithm *PR*.

## 7  Conclusion

We presented learning objectives and algorithms for stochastic structured prediction under bandit feedback. The presented methods "banditize" well-known approaches to probabilistic structured prediction such as expected loss minimization, pairwise preference ranking, and cross-entropy minimization. We presented a comparison of practical convergence criteria based on early stopping with theoretically motivated convergence criteria based on the squared gradient norm. Our experimental results showed fastest convergence speed under both criteria for pairwise preference learning. Our numerical evaluation showed smallest variance for pairwise preference learning, which possibly explains fastest convergence despite the underlying non-convex objective. Furthermore, since this algorithm requires only easily obtainable relative preference feedback for learning, it is an attractive choice for practical interactive learning scenarios.

**Acknowledgments.**

This research was supported in part by the German research foundation (DFG), and in part by a research cooperation grant with the Amazon Development Center Germany.

## Footnotes

[2]See [2] for an overview of bandit learning from consistent and inconsistent pairwise comparisons.

[3]For constant learning rates, [21] show even faster convergence in the search phase of strongly-convex stochastic optimization.

[4]For example, these constants appear as $\mathcal{O}\left(\frac{L}{\epsilon} + \frac{L\sigma^2}{\epsilon^2}\right)$ in the complexity bound for non-convex stochastic optimization of [7].

[5]Note that the squared gradient norm upper bounds the suboptimality criterion s.t. $\|\nabla J(w_t)\|^2 \geq 2\lambda J(w_t)] - J(w^*)$ for strongly convex functions. Together with the use of constant learning rates this means that we measure convergence to a point near an optimum for strongly convex objectives.

[6]http://www.statmt.org/wmt07/shared-task.html

[7]`http://www.cnts.ua.ac.be/conll2000/chunking/`

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
