[Reviews · NeurIPS 2016]

Reviewer 1

Summary

This paper proposes a stochastic online learning method for the task of structured prediction. In this setting, the learner doest not get the correct structured output during training. Instead, it only gets bandit feedback from the labeler. The paper first proposes an online learning algorithm that learns model parameters via stochastic gradient descent; generalizes the learning method to pair-wise comparison of structured outputs; provides an optimization approach with Cross-Entropy Minimization; and theoretically analyzes the convergence property of the optimization approach.

Qualitative Assessment

Summary: This paper proposes a stochastic online learning method for the task of structured prediction. In this setting, the learner doest not get the correct structured output during training. Instead, it only gets bandit feedback from the labeler. The paper first proposes an online learning algorithm that learns model parameters via stochastic gradient descent; generalizes the learning method to pair-wise comparison of structured outputs; provides an optimization approach with Cross-Entropy Minimization; and theoretically analyzes the convergence property of the optimization approach. Pros: The paper proposes an online stochastic learning algorithm for minimizing the expected loss of structured predictions; gives a method of learning from pair-wise comparisons; and theoretical analyze the convergence rate. Cons and Detailed Comments: Algorithm 1 is a straightforward stochastic optimization method for the minimization of Equation (2). The contribution of algorithm 1 is not significant. The probability of pairs are not well defined. First, the model fails to define relation with (partial) order among labels. (quote: "Stronger assumptions on the learned ranking can be made if asymmetry and transitivity of the observed ordering of pairs is required"). Thus, there is some probability to get y1 > y2, y2 > y3, and y1 < y3. I think this is not acceptable. Second, the distribution of label pairs not only defines the relation between labels in a pair, but also defines how a pair is sampled. This distribution is defined by the model parameters. However, the model parameters should only define relations among labels, but not how label pairs are generated. The writing of the paper can be greatly improved. The paper does not connect very well with the existing literature. For example, the section "Related Work" includes MDPs and bandit learning. However, I don't see the direct relation between any of the two sub-fields and the work. The introduction section mentions the principle of "exploration/exploitation", however, I don't see how this principle is applied to the proposed method. In the experiments section, three methods are compared. It seems all three methods are proposed in this work (or the paper should be more clear about the three methods.) One specific question: why no baseline methods are included? No sensitivity analysis for meta parameters. The standard deviations in Table 2 which are averaged over only 3 runs are unusually small. Is there a reason for this? The number of feedback iterations makes it untenable for a human expert to be involved. I don't think the bandit feedback (as per the problem setup) is practically viable -- It is a lot of human effort to repeatedly label for the same structured input. In that sense, existing approaches like Co-Active Learning are much better: see https://www.jair.org/media/4539/live-4539-8673-jair.pdf

Confidence in this Review

3-Expert (read the paper in detail, know the area, quite certain of my opinion)


Reviewer 2

Summary

This paper presents a probabilistic structured prediction algorithm for learning from bandit feedback. At each step, the model samples an output structure and obtain the corresponding loss. Then an unbiased update is conducted to update the model. The authors describe three variances of the algorithm based on different losses. Convergence analysis is conducted based on [9]. Experiment results on two real-world data verify the empirical performance of the algorithms.

Qualitative Assessment

Technical Quality: Overall, the quality is okay. The paper presents both convergence analysis and empirical performance on two real applications. Please see more comments below. Novelty/originality: The key idea of the paper follows [20]. However, the paper presents an additional iteration complexity analysis and applied with two other losses. One additional experiment set (chunking, a sequential tagging task) is used in this paper. Potential impact or usefulness Structured prediction with bandit feedback seems an interesting topic to me. The proposed algorithm extends [20] and is pretty general. It can be potentially applied in other applications. Clarity and presentation Overall, the paper is well-written and motivated well. Please see some minor comments below. Other comments: - It seems to me the iteration complexity also depends on the size of output space and the structure of the loss, \Delta (y). Specifically, the iteration complexity may be exponentially large in the size of y. For example, if there is only one instance (x,y), and loss is 0-1 loss (only one gold y has loss 0 and others have loss 1.). In such case, the algorithm may take an exponentially large number of iterations to find the right y in order to make a right update of the model. Could the authors comment on it? - Joint prediction with bandit feedback has also been discussed in [a]. - The experimental results do not agree with the analysis. PR is non-convex and CE is convex. However, CE requires much more iterations to converge. - Line 173: it is unclear in the manuscript why s_t is an unbiased estimation of \nabla J(w_t). Perhaps, the authors can elaborate more if the space allowed. -It seems to me, the approach EL is applying a policy gradient method on the risk minimization formulation in [b]. I would like to see more discussion about the relations between the proposed approach and policy gradient methods. - Perhaps, it is useful to present the learning curves of ER, PR, CE. - What is the relationship between PR and the duel bandit algorithm in [20]? - Perhaps, the authors can move some descriptions in Section 1 to Section 5. They are too detailed. [a] learning to search better than your teacher [b]Training Log-Linear Models with Cost Functions.

Confidence in this Review

2-Confident (read it all; understood it all reasonably well)


Reviewer 3

Summary

The authors present a pairwise preference learning algorithm in a bandit framework for stochastic structured prediction. The framework uses Gibbs distribution for output structure sampling which trades off between exploration vs. exploitation, and receives a partial feedback to update the model using stochastic gradient descent. The gradient is further derived from three different approaches; Expected Loss minimization, convexified Cross Entropy minimization, and the proposed pairwise Preference Learning. The experiments show faster convergence and smaller variance for the proposed approach.

Qualitative Assessment

The main contribution of the paper is developed from [20] where the expected loss minimization algorithm is replaced by two other objectives. Having a fixed learning framework and changing the objectives, i.e. the gradients, is an interesting idea and is useful for comparisons. However, it is also interesting to see out of the box of bandits. The focus of the paper is on bandit based algorithm, and the authors state that the partial feedback is sufficient for structured prediction, while there is no proof either experimental or theoretical that supports the claim. The storyline of the paper could have been written more distinctively. The section on related work misses somewhat literature in structured prediction as well as the relation of their contribution to the core methods used in the next sections. In addition, the performance of different methods in the “chunking” dataset is not well studied. The results show that the pairwise preference learning does not outperform the baseline method, while they converge in the same order (smaller factor). It would be helpful to better understand the advantages and the nature of the proposed method. Moreover, plotting the converging behavior over time would be more informative rather than in a fixed point. Overall, the idea is valuable to be introduced. Nevertheless, adding some non-bandit baselines combined with a concise presentation would improve the manuscript.

Confidence in this Review

2-Confident (read it all; understood it all reasonably well)


Reviewer 4

Summary

This paper presents algorithms for learning a structured classifier in the bandit setting, where the reference structure is not available to the learner and, instead, the predicted structures can be evaluated to produce a loss. The paper gives a convergence analysis of the algorithms and evaluates them on machine translation and noun phrase chunking experiments.

Qualitative Assessment

The paper is reasonably well written, and the main point comes through. The experiments illustrate the effectiveness of the proposed methods. A couple of places require clarifications: - I don't see how the second equality in Eq 7 can be true. Is this an assumption? - It is not clear how Eq 10 represents the stochastic gradient of the objective. Should the y_t inside the expectation not have the tilde? Also, in a few places in the explanation following the equation. How do the proposed algorithms compare to Shivaswamy & Joachims 2012, "Online Structured Prediction via Coactive Learning"? Their learning goal is similar to the presented work, namely to learn a structured classifier with only user feedback. The idea of learning in the bandit setting has also been explored in the semantic parsing community, though without a formal analysis. The setting is that the learner does not observe a reference semantic parse, but only gets feedback as to whether the parse produced the correct result. Some representative work: - Clarke, Goldwasser, Chang, and Roth, 2010. "Driving semantic parsing from the world's response." - Liang, Jordan and Klein, 2011. "Learning Dependency-Based Compositional Semantics"

Confidence in this Review

2-Confident (read it all; understood it all reasonably well)


Reviewer 5

Summary

The main aim of this paper is to consider a class of minimum-loss classifiers and replace the proper loss by bandit feedback. I was initially puzzled by the definition of expected loss given in equation 1 since it doesn’t follow the usual MAP or minimum-loss decision rules. However, given that the prediction here is made in terms of sampling, i.e., ybar ~ p(y|x), the definition is given correctly. For p(y|x) \in the exponential family, this expected loss is easily differentiable. After showing the standard gradient for this class in equation 2, the paper proposes three alternative formulations based on feedback: 1. an expected loss minimization (most similar to the original based on the proper loss), labelled EL; 2. a minimization of pairwise label disagreement, labelled PR ("pairwise preference") 3. a minimization of an n-best cross-entropy, labelled CE. Algorithm 1 shows the approach very clearly: with an exploitation/exploration strategy, the algorithm samples a label from the current model, obtains feedback for it, and adjusts the model based on the label gradient. Section 5 shows a convergence proof. Section 6 reports on NLP experiments on automated translation (WMT 2007 dataset) and noun-phrase chunking (CoNNL-2000 dataset). Bandit feedback is, as often, simulated from ground truth.

Qualitative Assessment

In general, I like this paper. It adds to the growing field of bandit feedback papers with a clear contribution: three original, alternative ways of computing the gradient in a simple sample/adjust learning algorithm. This paper is an extension of its reference [20], in my opinion, significant for publication. The bottom line of the experimental results is that the proposed approaches achieve nearly (nearly) the same accuracy as the corresponding methods with proper loss ("full information" column). A proof of convergence is also given. These are my remarks: 1. Minor: the paper refers to "structured prediction", but the proposed approach has no relation to structure and can be applied to single, i.i.d. outputs as well. In other words, y may be structured or not, without any relevance to the proposed approach. 2. Medium: The experimental results only compare the proposed approaches with the corresponding methods with proper loss. However, the state of the art on CoNNL-2000 chunking is significantly better than this (over 95% F1 score); there should be some comments about this (e.g., the prediction model is unimodal, log-linear or else; advantages over other methods).

Confidence in this Review

2-Confident (read it all; understood it all reasonably well)


Reviewer 6

Summary

This paper applies stochastic approximation methods in training log linear probabilistic structured prediction models under the bandit setting. It differs from the full information setting in that the learner only receives partial feedback for every prediction in the form of task loss evaluation, rather than a ground truth output with which to validate its prediction. The author(s)' stated main contribution of the paper is to apply stochastic first order (SFO) methods to learning with pairwise comparison feedback, and showing that it achieves the fastest convergence over two other settings (loss and cross-entropy minimization) based on empirical evaluations. The convergence properties of all three methods are analyzed based on existing results in the literature, which assume smoothness and boundedness conditions. Despite worst-case theoretical guarantees that favor the strongly convex cross-entropy objective, the author(s) reason that pairwise learning performed better due to smaller constant factors such as the variance of stochastic gradient updates. This conclusion is drawn from experimental evaluations.

Qualitative Assessment

A main result of this paper is that under bandit feedback, applying SFO methods in pairwise preference ranking (PR) and expected loss minimization (EL) can lead to better empirical convergence rates than in cross-entropy (CE) minimization, despite theoretical results that might suggest otherwise. The author(s) explained this result by estimating the constant factors of the bounds, such as the variance of stochastic updates, showing that they are orders of magnitude smaller for PR and CE. This leads to a statement in the conclusion that '... constants such as the variance of the stochastic gradient update can offset possible theoretical gains in convergence speed.' Applying SFO methods in PR is interesting and has potential applications, but I see several issues with drawing the conclusion above from empirical evaluations, 1. The learning rates for all three methods were fixed at 1e^-6: the authors stated in line 183-184 that theoretical convergence for strongly convex functions (which applies to CE) were analyzed under diminishing learning rates, but used a constant rate in the experiments. I am not aware of similar bounds that hold under constant learning rate in the strongly convex case. How can one be certain that the apparent advantages of PR and EL over CE are not due to suboptimal choices of learning rates? 2. There is no justification of why 1e^-6 is chosen. Does the conclusion still hold under different values? 3. The constant factor estimates are based on averaging the results of 3 runs of each algorithm; I think the sample may be too small. Another issue of this paper is that a number statements were presented without proper justification or context, which makes it difficult to assess their correctness. This is problematic because several of them are crucial for the paper's main results to hold. For example, 1. line 158-159, 'Note that multiplying the gradient by ...' -- Isn't this multiplication by 1? It is not clear how the sampling is done. 2. line 159-162, 'While minimization of this objective ... This further increases the variance ...' -- This explanation is unclear to me. Since this relates to a central claim of the paper, it would be helpful to clarify it. 3. line 175-177 -- These assumptions are crucial and it would help to be more precise here, e.g., by deriving the bounds on the variance explicitly under all 3 objectives. 4. line 187-189, For constant learning rates, even faster convergence can be shown ... -- What is the implication of this and how is 'faster' characterized? 5. line 193-194, 'Second, the standard asymptotic complexity bounds hide the constants ...' -- It would be helpful to state the bounds precisely with the inclusion of epsilon and all these constants, along with any other factors that may affect convergence rates (e.g, dimension of w and learning rate choices). It would also help to provide more context in Related Work section. Instead of generic references to reinforcement learning and bandit learning (which are too broad), the author(s) could survey existing approaches to probabilistic structured prediction, going into more depth on how SA methods have been applied to closely related problems, and what this paper does differently. Overall, I find the evidence presented in this paper and its explanations inadequate to support the conclusions. The paper could also use improvements in clarity and rigor of writing, especially when presenting theoretical results of convergence rates and the factors that influence them. For these reasons, I am unable to recommend it for publication.

Confidence in this Review

2-Confident (read it all; understood it all reasonably well)